# Detection of Green Asparagus in Complex Environments Based on the Improved YOLOv5 Algorithm

**DOI:** 10.3390/s23031562

**Published:** 2023-02-01

**Authors:** Weiwei Hong, Zenghong Ma, Bingliang Ye, Gaohong Yu, Tao Tang, Mingfeng Zheng

**Affiliations:** 1Faculty of Mechanical Engineering, Zhejiang Sci-Tech University, Hangzhou 310018, China; 2Special Equipment Institute, Hangzhou Vocational & Technical College, Hangzhou 310018, China; 3Key Laboratory of Transplanting Equipment and Technology of Zhejiang Province, Hangzhou 310018, China

**Keywords:** green asparagus, detection, improved YOLOv5, coordinate attention, BiFPN

## Abstract

An improved YOLOv5 algorithm for the efficient recognition and detection of asparagus with a high accuracy in complex environments was proposed in this study to realize the intelligent machine harvesting of green asparagus. The coordinate attention (CA) mechanism was added to the backbone feature extraction network, which focused more attention on the growth characteristics of asparagus. In the neck part of the algorithm, PANet was replaced with BiFPN, which enhanced the feature propagation and reuse. At the same time, a dataset of asparagus in complex environments under different weather conditions was constructed, and the performance variations of the models with distinct attention mechanisms and feature fusion networks were compared through experiments. Experimental results showed that the mAP_@0.5_ of the improved YOLOv5 model increased by 4.22% and reached 98.69%, compared with the YOLOv5 prototype network. Thus, the improved YOLOv5 algorithm can effectively detect asparagus and provide technical support for intelligent machine harvesting of asparagus in different weather conditions and complex environments.

## 1. Introduction

Asparagus is recognized as a healthy vegetable with high nutritional value, ranking among the world’s “Top 10” famous vegetables and having the reputation of being the king of vegetables in the international market [1,2]. In 2020, China’s asparagus planting area and output ranked first in the world, with 1.501 million ha and 8.613 million t, accounting for 90.6% and 88.1% of the world’s total, respectively [3]. The tender stems of asparagus are its edible parts. However, considerable manual labor is required for selective harvesting because of the inconsistency in the growth direction and maturity of the tender shoots. Asparagus production is currently facing a series of problems, such as high labor intensity, high operating cost, low degree of mechanization, and low production efficiency, which seriously restrict the sustainable development of the asparagus industry [4]. Therefore, the realization of intelligent machine harvesting of asparagus has become an urgent need to promote its industrial development.

The recognition and detection technology of asparagus is considered the key technology to realize the intelligent machine harvesting of asparagus. Sakai et al. [5] used the technology of a laser sensor to locate and detect asparagus, and the recognition success rate was 75%. Peebles et al. [6,7] compared the sensor technologies used for asparagus harvesting and investigated the method used for determining the position of the ridge surface under the asparagus harvesting scenario. Kennedy et al. [8] proposed the concept of perceptual channels based on multiple cameras to achieve the localization of green asparagus. Peebles et al. [9] compared the detection effects of Faster-RCNN and SSD algorithms for asparagus and determined that Faster-RCNN can obtain an F1 value of 0.73, which is more suitable for asparagus detection. Leu et al. [10] developed a robot for the selective harvesting of green asparagus, which uses RGBD cameras to obtain 3D information on asparagus and ridge surface and realizes asparagus localization through clustering. To date, only a few studies on the recognition and detection of asparagus have been conducted worldwide, and two major problems in this field must be addressed: first, the success rate of asparagus recognition is low, the model algorithm needs to be further investigated, and the technical scheme of recognition and detection must be improved. Second, the current research on asparagus recognition and detection is aimed at the simple scenario of asparagus in spring (i.e., asparagus has no stalk in that season) but has not been applied to scenarios for complex environments in summer and autumn. In summer and autumn, the stems and leaves of asparagus are clustered, and the planting environment is more complicated. The shape and color of asparagus shoots and stems are similar, and numerous stacking conditions are involved. Asparagus sprouts are similar in color to leaves and weeds, and many leaves and weeds are obscured or fade into the background. Moreover, asparagus occupies a small proportion of the whole scene, so the detection becomes small target detection. Therefore, the recognition and detection technology of asparagus is difficult to develop.

In recent years, research on the visual recognition and detection technology of crops has received considerable attention from scholars across the world [11,12]. Related results have been applied in agricultural production and can provide meaningful references and solutions for asparagus recognition and detection. YOLO series [13,14,15,16] algorithms have been applied to the recognition and detection of apples, citrus, winter jujubes, bananas, popular teas, cucumber, corn seedlings, and flower buds, as well as pests and diseases, by many scholars worldwide because of its advantages of fast detection speed and high accuracy. Zhao et al. [17] used the YOLOv3 deep convolutional neural network to enable apple pickers to identify and locate fruits under various conditions, such as occlusion, adhesion, and bagging under varying lighting conditions, around the clock. Lv et al. [18] proposed an identification method of apple growth shape based on the improved YOLOv5 algorithm to facilitate the harvesting robot to adopt different harvesting attitudes. Xiong et al. [19] proposed the multiscale convolutional neural network Des-YOLOv3, which realized the recognition and detection of ripe citrus in a complex environment at night. Liu et al. [20] proposed a recognition method based on the improved YOLOv3, which realized fast and accurate recognition of winter jujube in natural scenarios. Fu et al. [21] applied YOLOv4 to recognize banana bunches and stems in natural environments. Xu et al. [22] compared the target detection effect of different backbone feature extraction networks based on YOLOv3 and selected DenseNet201 as the backbone feature to extract the network to realize the recognition and detection of popular teas. Bai et al. [23] integrated the U-Net, YOLO, and SSD networks to achieve the target detection of cucumbers in complex natural environments. Quan et al. [24] used the YOLOv4 convolutional neural network to achieve accurate identification of corn seedlings and farmland weeds in complex field environments. Li et al. [25] used the improved YOLOv4 to realize the recognition of kiwifruit flowers and buds in preparation for automatic machine pollination. Qi et al. [26] proposed an improved YOLOv5 algorithm based on the attention mechanism to realize the detection of tomato pests and diseases. Zhang et al. [27] added ECA, hard-swish activation function, and focal loss function into YOLOX to detect cotton diseases and pets. Zhang et al. [28] proposed a new method of CBAM + ASFF-YOLOXs for guiding agronomic operation based on the identification of key growth stages of lettuce. Nan et al. [29] used NSGA-II-based pruned YOLOv5l to detect green pepper quickly in the field environment. Xu et al. [30] improved the citrus fruit detection accuracy for a picking robot in complex scenes and enables real-time picking operations by proposing a novel detection method called high-precision and lightweight YOLOv4. Fan et al. [31] realized real-time defects detection for apple sorting using NIR cameras with a pruning-based YOLOv4 network. Thus far, no research on the YOLO algorithm in asparagus recognition and detection has been reported worldwide, and the recognition accuracy of the algorithm must be further improved and perfected according to different application scenarios.

In summary, this study proposed an improved YOLOv5 algorithm for the recognition and detection of green asparagus in complex environments. The coordinate attention (CA) mechanism was added to the backbone feature extraction network [32]. Such improvements can enhance the feature learning capability and strengthen the attention to the feature information of asparagus. The PANet was modified to BiFPN [33,34,35], which can strengthen the fusion of features, reduce the loss of feature information, and improve the detection effect of the model. Meanwhile, comparative experiments and verification of different algorithms were conducted.

## 2. Materials and Methods

### 2.1. Image Data Collection

The photos were obtained from a side angle under different lighting conditions, as shown in Figure 1. Specifically, 701 photos were collected under sunny conditions with strong sunlight on 31 May 2022, and 601 photos were collected under cloudy conditions with low light on 3 June 2022, at the location of the asparagus planting base of Hangzhou Jiahui Agricultural Development Co., Ltd. in Hangzhou, China. The asparagus greenhouse is about 6 × 60 m with 4 rows. The dimensions of each row are shown in Figure 1. The camera used for data acquisition was SONY ILCE-6100. It was fixed on a tripod and manually controlled for image acquisition. The camera setting parameters are shown in Table 1, and the photos are shown in Figure 2. The aforementioned figure shows that the density of asparagus growth is inconsistent. The growth direction is irregular, the branches and leaves are severely blocked, the color of the asparagus sprouts is the same as that of weeds and asparagus stems and leaves, and the shape of the asparagus sprouts is similar to that of asparagus stems and stacked on each other. All these situations cause considerable difficulty in identification.

### 2.2. Data Labeling

LabelImg software was used for the labeling process, and the labeled files were all in XML format. When applied to the YOLO algorithm, the labeled files only need to perform format conversion. Given that the machine only harvests the current row during the harvesting process, it only labeled the asparagus where the current row was located and considered the asparagus in the distance as the background. The labeled images are shown in Figure 3.

### 2.3. Data Enhancement

A total of 1302 green asparagus photos were labeled. These photos were enhanced to increase the number of samples to improve the adaptability and robustness of the model. Given the uprightness of asparagus during the harvesting process, the photos were flipped horizontally, and a total of 2604 pictures were obtained, as shown in Figure 4. The training, validation, and test datasets were divided according to the ratio of 8:1:1. Specifically, 2084 photos were used for training, 260 photos were used for verification, and 260 photos were used for testing.

### 2.4. Improved YOLOv5 Network Model

#### 2.4.1. YOLOv5 Algorithm

YOLOv5 can be divided into three parts, namely, backbone, FPN, and YOLO Head, as shown in Figure 5. YOLOv5 uses CSPDarknet as the backbone feature extraction network, and the input image depicts features extracted using CSPDarknet to obtain three effective feature layers. YOLOv5 uses PANet to achieve enhanced feature extraction of images, fuses the three effective feature layers obtained from the main part, and continues to extract features from the obtained effective feature layers, not only upsampling the features but also downsampling them to attain feature fusion. YOLO Head is a classifier and regressor of YOLOv5, which judges the corresponding objects of feature points according to the feature layer.

#### 2.4.2. Coordinate Attention Mechanism

The CA mechanism performs average pooling in the horizontal and vertical directions and fuses spatial information in a weighted manner through spatial information encoding. The module of CA is shown in Figure 6.

First, the CA mechanism averages pools horizontally and vertically for a given input *x*.
(1)$$Z_{c}^{h}\left(h\right)=\frac{1}{W}\underset{0\leq i<W}{\sum }x_{c}\left(h,i\right),$$
(2)$$Z_{c}^{w}\left(w\right)=\frac{1}{H}\underset{0\leq i<H}{\sum }x_{c}\left(j,w\right),$$
where $Z_{c}^{h}\left(h\right)$ is the output of the *c*th channel with height *h*, and $Z_{c}^{w}\left(w\right)$ is the output of the *c*th channel with width w.

Second, the CA mechanism concatenates and compresses the channels via a convolution transform to change the number of channels from C to C/r, where r is used to control the reduction rate. The spatial information in the vertical and horizontal directions is encoded through batch normalization and non-linear processing.
(3)$$f=\delta \left(F_{1}\left(\left[z^{h},z^{w}\right]\right)\right),$$
where *F*_1_ is the 1 × 1 convolution transform, [∙,∙] is the splicing operation, and *δ* is the non-linear activation function.

Then, f is decomposed into two separate tensors $f_{h}$ and $f_{w}$, and two 1 × 1 convolution transforms *F_h_* and *F_w_* are used to obtain the same number of channels as the input *x*.
(4)$$g^{h}=\sigma \left(F_{h}\left(f^{h}\right)\right),$$
(5)$$g^{w}=\sigma \left(F_{w}\left(f^{w}\right)\right),$$
where *F_h_* and *F_w_* are the 1 × 1 convolution transforms, and *σ* is the sigmoid function.

Finally, the output y of the CA attention module is computed.
(6)$$y_{c}\left(i,j\right)=x_{c}\left(i,j\right)\times g_{c}^{h}\left(i\right)\times g_{c}^{w}\left(j\right)$$

#### 2.4.3. BiFPN Network Structure

The mechanism of the neck part of the YOLOv5 algorithm is a combination of FPN and PAN, which aggregates parameters from different backbone layers to various detection layers. The structure is shown in Figure 7b. Although PANet effectively strengthens the feature fusion capability of the network, the input of the PAN structure is all feature information processed by the FPN structure, and the lack of the original feature information of the backbone feature extraction network will easily lead to deviations in training and learning, which will affect the detection accuracy. Accordingly, PANet is replaced with BiFPN in the model. BiFPN simplifies the structure of FPN + PAN and removes the node with only one input edge and one output edge. If the input and output nodes belong to the same layer, then an extra edge will be added. The structure is shown in Figure 7c.

#### 2.4.4. Improved Algorithm of YOLOv5-CB

In summer and autumn, the growth environment of asparagus is complex, its growth density is inconsistent, and its growth direction is irregular. The color of asparagus sprouts is the same as those of stems, leaves, and weeds, and their shape is similar to that of asparagus stems. The identification and detection of asparagus are difficult due to these various factors. To improve the accuracy of the detection algorithm based on the YOLOv5m model, this study added the CA mechanism to the backbone feature extraction network, which enhanced the weight and representation of the target of interest and ensured the effective extraction of small target features. In the neck part, PANet was replaced with BiFPN, which achieved higher-level feature fusion and further improved the accuracy of the algorithm. The improved algorithm is called YOLOv5-CB (CA + BiFPN), and its network structure is shown in Figure 8.

### 2.5. Training Environment and Evaluation Indicators

The network was established by PyCharm with the deep learning framework PyTorch1.9.0. The experiment was run on Windows 10, with an Intel^®^ Core™ i5-6500 CPU at basic frequency 3.20 GHz, 24 GB RAM, an Nvidia GeForce RTX 3060 Super graphics card, and accelerated by CUDA 11.1 and CUDNN 8.0.5.

All images were adjusted to 640 × 640 pixels to meet the input requirements of the model, the batch size corresponding to the computer hardware was set to 16, and the network was optimized by the SGD optimizer. The settings of the other hyperparameters are shown in Table 2. To reduce the training time, the transfer learning method was adopted, and the officially provided pretraining weights were loaded to start the training of the model.

Detection accuracy and detection speed are important indicators for measuring the model performance. The indicators of detection accuracy include precision rate (*P*), recall (*R*), average precision (*AP*), and mean average precision (*mAP*). The indicator of model detection speed uses the number of detected image frames per second (*FPS*). The calculation formulas are expressed in Equations (7)–(11):(7)$$P=\frac{TP}{TP+FP}\times 100\%,$$
(8)$$R=\frac{TP}{TP+FN}\times 100\%,$$
(9)$$AP=\int_{0}^{1}P\left(R\right)dR,$$
(10)$$mAP=\frac{1}{n}\overset{n}{\underset{i=1}{\sum }}AP_{i},$$
(11)$$FPS=\frac{N}{t},$$
where *TP* is the number of correctly predicted positive samples, *FP* is the number of falsely predicted positive samples, *FN* is the number of wrongly predicted negative samples, *n* is the number of target categories to be detected, *AP_i_* is the AP of the *i*th target class, *N* is the number of images to be detected, and t is the detection time.

## 3. Results

### 3.1. Results of the Improved Backbone

To verify the effectiveness of the attention mechanism and explore the detection effect of different attention mechanisms embedded in the algorithm, the three commonly used attention mechanisms, namely, SE, ECA, and CBAM, were embedded in the backbone as the CA mechanism, and no modifications were made to the other parts. The modified models, namely, YOLOv5-CA, YOLOv5-SE, YOLOv5-ECA, and YOLOv5-CBAM, were tested and compared on the asparagus dataset. The experimental results are shown in Table 3.

Table 3 shows that, after adding the attention mechanisms, namely, CA, SE, ECA, and CBAM, to YOLOv5, mAP_@0.5_, mAP_@0.75_, and mAP_@0.5:0.95_ were all improved. In terms of detection speed, all the weight files became larger, which resulted in a small reduction in the detection speed. However, the network integrated with the CA mechanism exhibited less reduction in detection speed and considerable improvement in detection accuracy.

### 3.2. Results of the Improved Neck

To verify the effectiveness of the improvement of the neck part, this study changed the PANet structure in the original YOLOv5 algorithm to the BiFPN structure proposed in this study, and the rest remained unchanged. The improved algorithm, called YOLOv5-B, was compared with the original YOLOv5 algorithm on the asparagus dataset. The experimental results are shown in Table 4.

Table 4 shows that mAP_@0.5_, mAP_@0.75_, and mAP_@0.5:0.95_ increased by 2.10%, 9.42%, and 9.68%, respectively, with a small increase in the calculation and parameter amounts of the algorithm, by improving the neck part. The detection speed was slightly reduced, and the accuracy was considerably improved, which proved the effectiveness of the improved algorithm.

### 3.3. Results of the Ablation Experiment

The improvement method proposed in this study is to add different attention mechanisms (i.e., CA, SE, ECA, and CBAM) and modify the feature fusion network to BiFPN to obtain YOLOv5-CB, YOLOv5-SB, YOLOv5-EB, and YOLOv5-CBB. The following ablation experiments were designed to verify the effectiveness of these modification methods: (1) based on the original YOLOv5 algorithm, only one improved method was added to verify the improvement effect of each improved method on the original algorithm. (2) The attention mechanism and BiFPN were freely combined to select the optimal detection model. The experiments were conducted on the asparagus dataset. The experimental results are shown in Table 5.

Table 5 shows that the performance of the algorithm was improved after the backbone and neck were improved. YOLOv5-EB achieved the best mAP_@0.5_ of 98.74%; and YOLOv5-CB achieved the best mAP_@0.75_ of 94.88% and the best mAP_@0.5:0.95_ of 82.94%. However, in terms of the detection effect and detection speed of the model, the performance of YOLOv5-CB is relatively balanced. The mAP_@0.5_, mAP_@0.75_, and mAP_@0.5:0.95_ of YOLOv5-CB were considerably improved compared with those of the original YOLOv5, and the reduction in detection speed was small. Therefore, YOLOv5-CB was selected to deal with the detection of asparagus in complex environments.

### 3.4. Results of the Comparative Experiment

To further prove the superiority of the algorithm proposed in this study, it was compared with the YOLOv5, YOLOv4, YOLOv3, and Faster-RCNN algorithms on the asparagus dataset. Figure 9 shows the change curve of mAP during training. The mAP_@0.5_ of Faster-RCNN, YOLOv3, YOLOv4, YOLOv5, and YOLOv5-CB can reach 90.67%, 91.55%, 92.09%, 92.74%, and 98.52%, respectively. The initial accuracy rate of YOLOv5-CB was low during training, and the accuracy rate fluctuated considerably. However, the convergence speed was high, and the accuracy rate was the highest. Figure 10 shows the comparison of the loss curves of YOLOv5 and YOLOv5-CB during the training process. The loss value of YOLOv5-CB is 0.015, which is 0.017 lower than that of the original YOLOv5, and the model is further optimized.

Faster-RCNN, YOLOv3, YOLOv4, YOLOv5, and YOLOv5-CB were verified on the test dataset. The experimental results are shown in Table 6. The YOLOv5 mosaic data enhancement can be achieved by splicing four images, as shown in Figure 11, which considerably enriches the background of the detected object. Table 6 shows that the performance of YOLOv5-CB has been further improved compared with that of YOLOv5. The mAP of YOLOv5-CB is 98.69%, which is 5.77%, 9.85%, 6.41%, and 4.22% higher than those of Faster-RCNN, YOLOv3, YOLOv4, and YOLOv5, respectively. The detection effect is shown in Figure 12 and Figure 13. The detection speed of YOLOv5-CB (i.e., 31 fps) is higher than those of Faster-RCNN, YOLOv3, and YOLOv4 (i.e., 11, 27, and 21 fps, respectively).

## 4. Discussion

This study proposed the YOLOv5-CB algorithm to detect asparagus in summer and autumn, whose environments are more complex. Due to the evolution of YOLO, the recognition precision of this algorithm increased by 11.88% to 97.88%, and the F1 value increased by 0.24 to 0.97 compared with Peebles’s study of spring asparagus recognition by Faster-RCNN. The recognition success rate increased by 22.88%, and the radar camera was replaced by the depth camera, compared with Sakai’s research. The recognition accuracy has been significantly improved. The cost of the machine can be reduced without requiring expensive equipment, such as laser sensors. Similarly, Zhang replaced the PANet with ASFF in the neck and added a CBAM attention mechanism to the backbone to realize the identification of key growth stages of lettuce. However, we still observed 24 cases of false or missed detection when testing on the test dataset. The analysis of these detection error scenarios showed that the main reasons for the identification errors are as follows: (1) serious stacking between asparagus sprouts, leaves, and stems occurred. (2) Asparagus sprouts or stalks were located at the edge of the image, and some parts were not included in the image. (3) Numerous leaves in the field of vision or background were detected. These conditions occur alone or together. The specific statistics are shown in Table 7.

The statistics showed that the first two main causes accounted for 87.1 percent of the total failures. First, the current algorithm has weak detection ability for dense small targets in complex environments. Second, the current algorithm mainly realizes the detection according to the characteristics of asparagus spears. For asparagus at the edge of the photo without spears, the current algorithm will lead to a high detection failure rate. In future research, the algorithm will be improved specifically to address these two main problems. For the main cause of serious stacking between asparagus sprouts, leaves, and stems, we will try to achieve iterative detection by changing the network structure and construct a new target detector VarifocalNet to efficiently detect dense targets. Aiming at the asparagus at the edge of the photo, we try to improve the feature extraction and fusion network so that the network can learn higher-level features and realize target detection through partial information.

## 5. Conclusions

This study added distinct attention mechanisms and modified the feature fusion network with BiFPN to achieve high accuracy and fast detection of asparagus in complex environments. To verify the effectiveness of the attention mechanism and BiFPN in the YOLOv5 model, this study compared the model performance after adding CA, SE, ECA, and CBAM, and modifying the feature fusion network to BiFPN. This study confirmed the optimal performance of the model after adding CA and modifying the feature fusion network to BiFPN. The mAP_@0.5_, mAP_@0.75_, and mAP_@0.5:0.95_ of the improved YOLOv5 model increased by 4.22%, 11.77%, and 11.50% and reached 98.69%, 94.88%, and 82.94%, respectively, compared with the YOLOv5 prototype network. The improved algorithm YOLOv5-CB also outperformed Faster-RCNN, YOLOv3, YOLOv4, and YOLOv5. The detection accuracy of the model was significantly improved. The improved YOLOv5-CB model showed its feasibility and superiority in application to the recognition and detection of asparagus in complex environments.

## Figures and Tables

**Figure 1 sensors-23-01562-f001:**
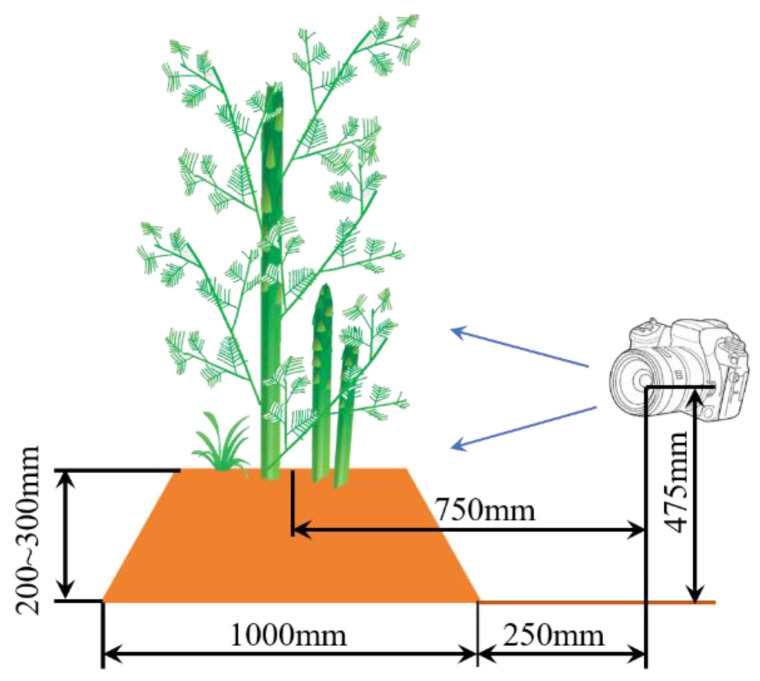
Image acquisition.

**Figure 2 sensors-23-01562-f002:**
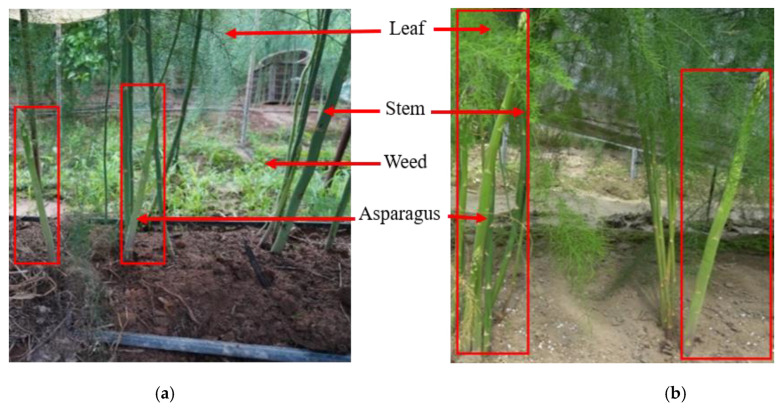
Photos of asparagus under different lighting conditions. (**a**) Image acquired on a cloudy day; (**b**) image acquired on a sunny day.

**Figure 3 sensors-23-01562-f003:**
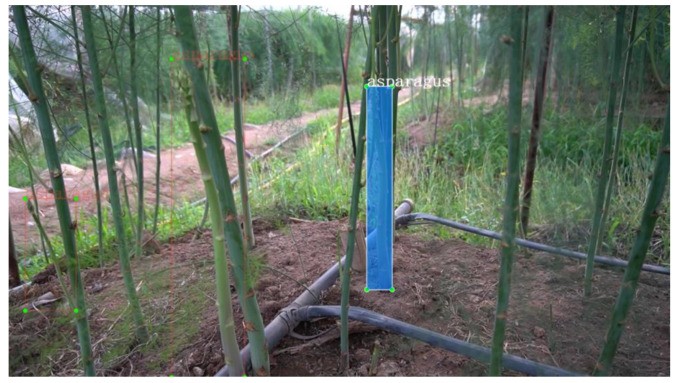
Labeling of the image.

**Figure 4 sensors-23-01562-f004:**
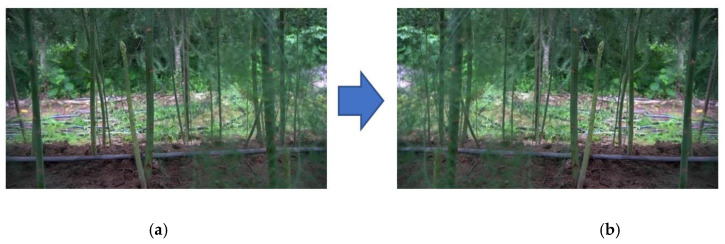
Photo after image enhancement. (**a**) Original photo; (**b**) photo after being flipped horizontally.

**Figure 5 sensors-23-01562-f005:**
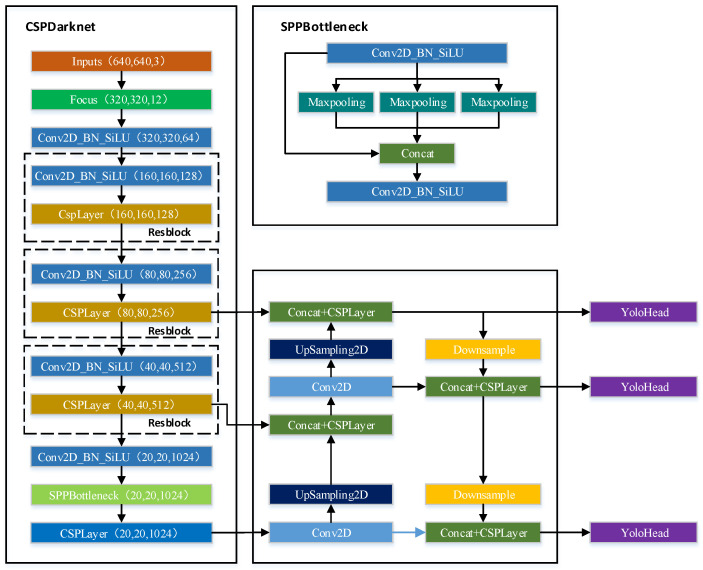
Model structure of YOLOv5.

**Figure 6 sensors-23-01562-f006:**
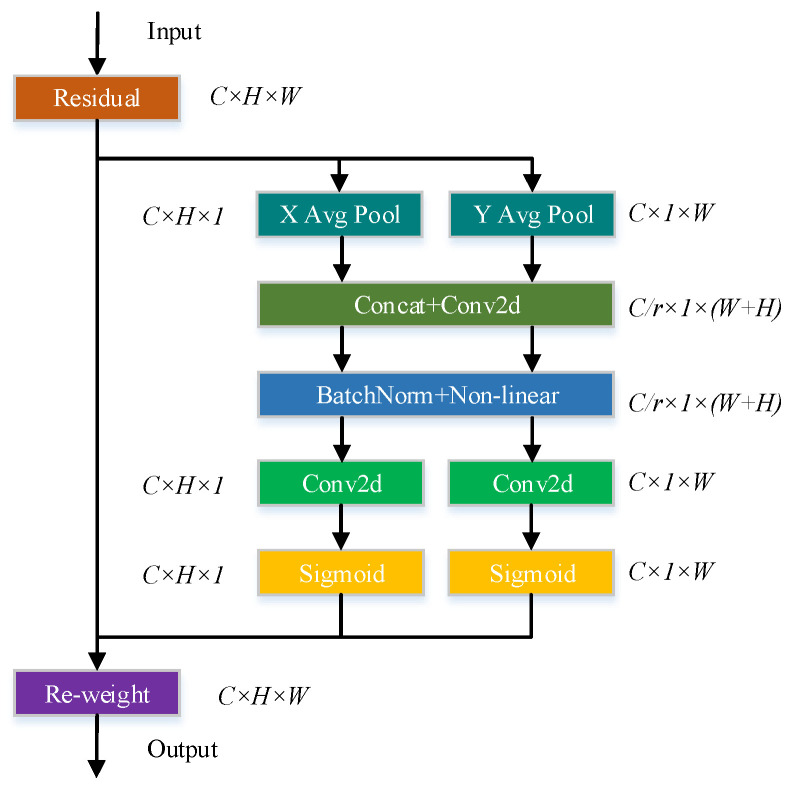
Module of CA.

**Figure 7 sensors-23-01562-f007:**
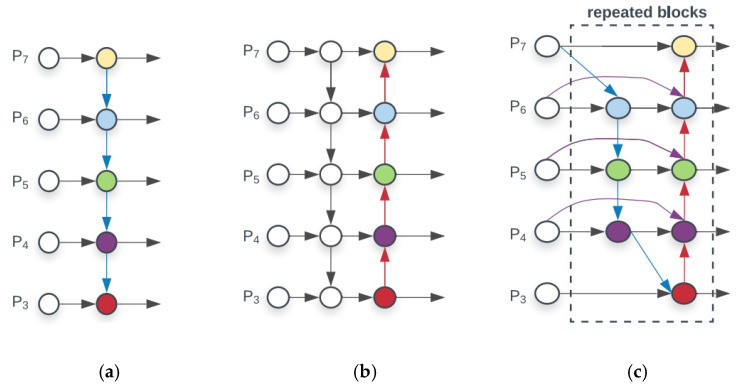
Schematic of the different feature fusion structures. (**a**) FPN; (**b**) PANet; and (**c**) BiFPN.

**Figure 8 sensors-23-01562-f008:**
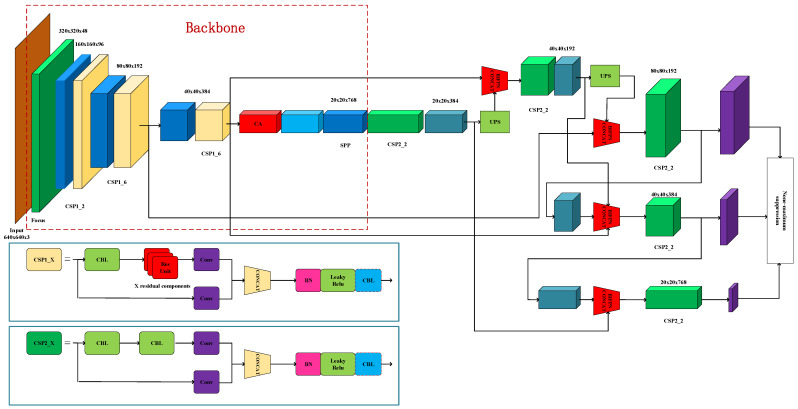
YOLOv5-CB network structure.

**Figure 9 sensors-23-01562-f009:**
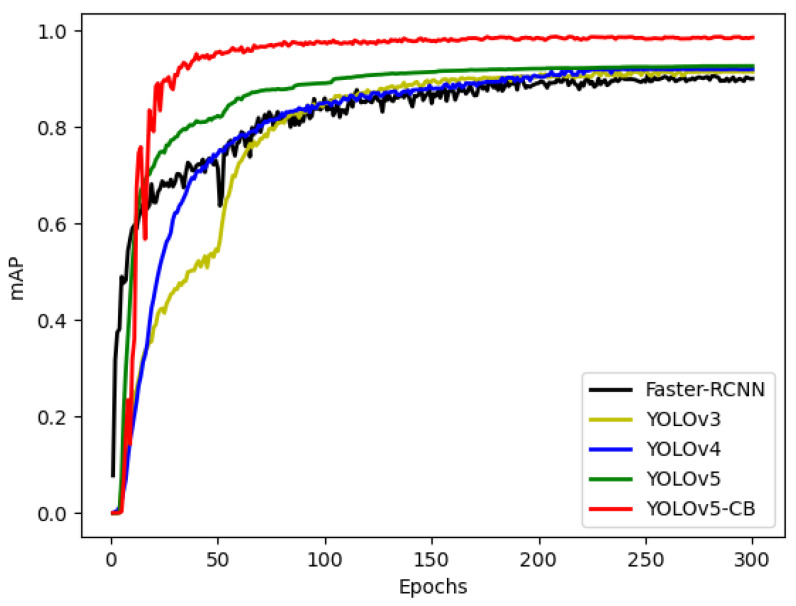
mAP change curve.

**Figure 10 sensors-23-01562-f010:**
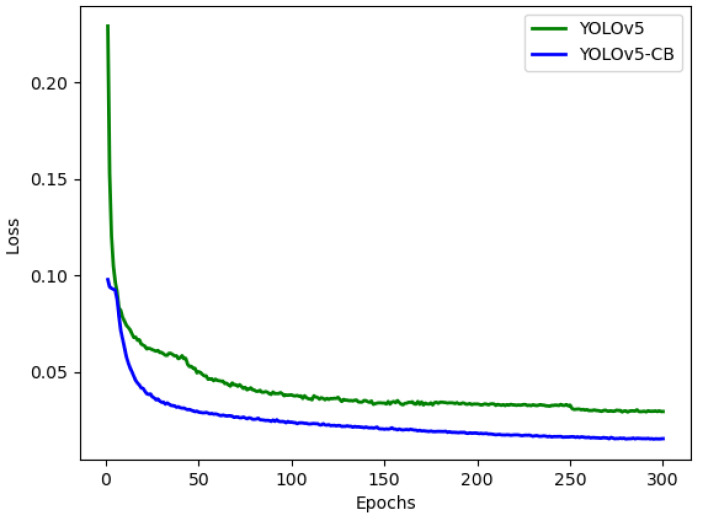
Loss curve.

**Figure 11 sensors-23-01562-f011:**
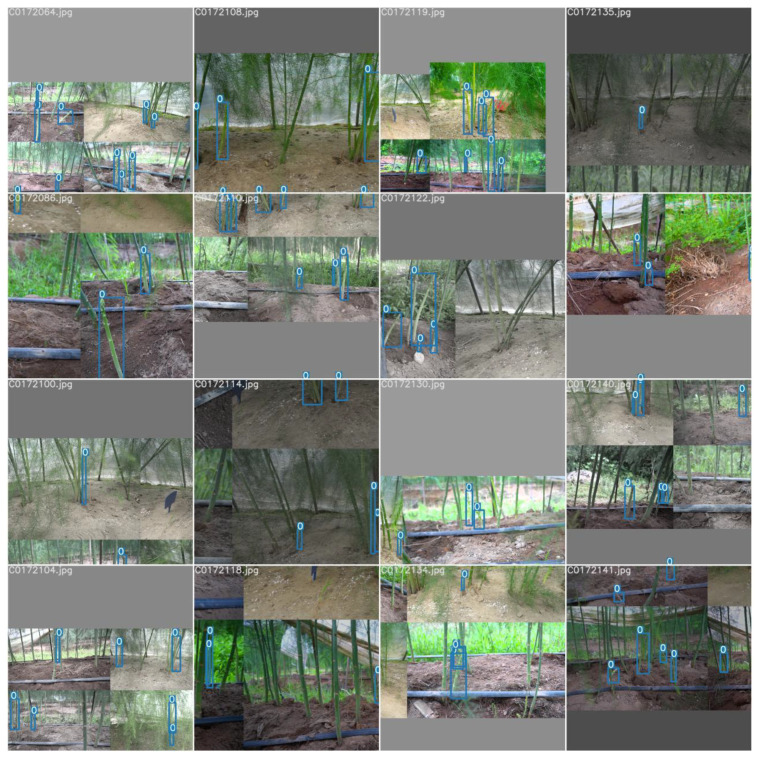
Mosaic data enhancement.

**Figure 12 sensors-23-01562-f012:**
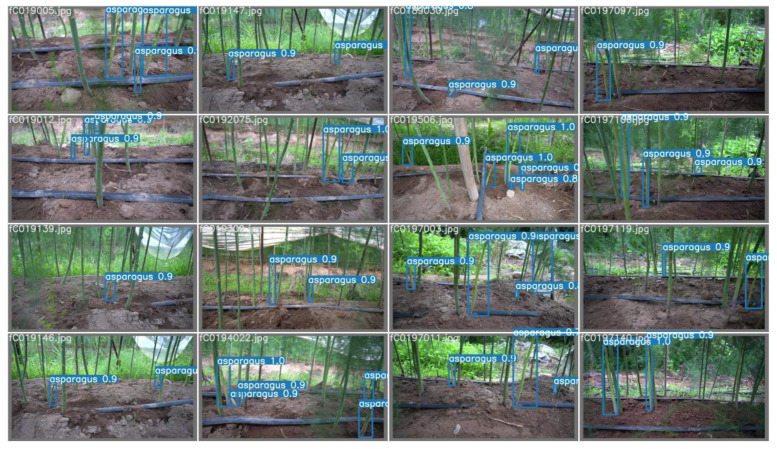
Batch detection effect.

**Figure 13 sensors-23-01562-f013:**
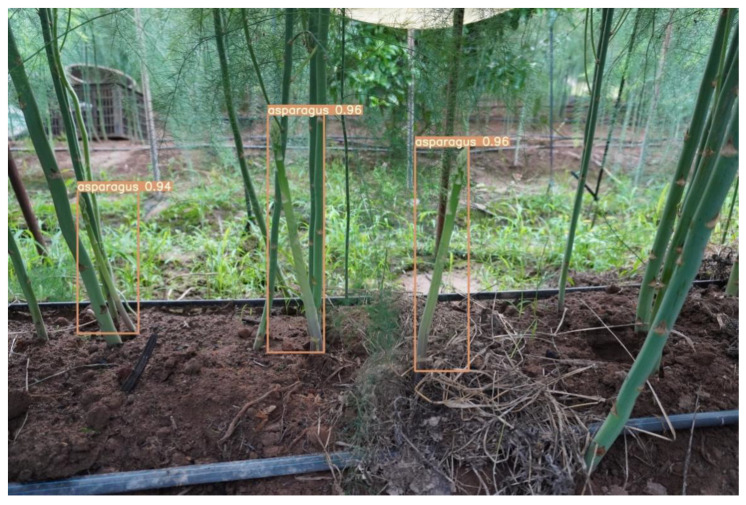
Detection effect.

**Table 1 sensors-23-01562-t001:** Camera setting parameters.

Variable	Value/State
Camera model	Sony ILCE-6100
Image size	1920 × 1080 pixels
Zoom	No zoom
Flash mode	No flash
Aperture av.	f/4; f/5
Exposure time av.	1/160 s;1/80 s
Focal length	16 mm
Operation mode	Manual
Macro	Off
Image type	JPG

**Table 2 sensors-23-01562-t002:** Training parameters.

Input Image Size	Batch Size	Momentum	Initial Learning Rate	Decay Index	Epochs
640 × 640	16	0.937	0.01	0.0005	300

**Table 3 sensors-23-01562-t003:** Results of integrating different attention modules.

Model	P (%)	R (%)	mAP_@0.5_	mAP_@0.75_	mAP_@0.5:0.95_	Weight Size (M)	Speed (fps)
YOLOv5	92.72	92.86	94.47	83.11	71.44	82.57	33
YOLOv5-CA	95.11	95.59	97.09	91.98	81.27	84.42	30
YOLOv5-SE	96.03	93.48	96.49	90.18	79.27	85.56	28
YOLOv5-ECA	95.32	95.51	96.69	91.49	80.83	84.35	30
YOLOv5-CBAM	95.43	93.79	96.30	88.11	77.94	83.31	32

P means precision rate; R means recall; mAP means mean average precision.

**Table 4 sensors-23-01562-t004:** Results of the improved neck part.

Model	P (%)	R (%)	mAP_@0.5_	mAP_@0.75_	mAP_@0.5:0.95_	Weight Size (M)	Speed (fps)
YOLOv5	92.72	92.86	94.47	83.11	71.44	82.57	33
YOLOv5-B	95.91	94.98	96.57	92.53	81.12	83.49	32

P means precision rate; R means recall; mAP means mean average precision.

**Table 5 sensors-23-01562-t005:** Results of the ablation experiment.

Model	P (%)	R (%)	mAP_@0.5_	mAP_@0.75_	mAP_@0.5:0.95_	Weight Size (M)	Speed (fps)
YOLOv5	92.72	92.86	94.47	83.11	71.44	82.57	33
YOLOv5-B	95.91	94.98	96.57	92.53	81.12	83.49	32
YOLOv5-CB	97.88	96.79	98.69	94.88	82.94	83.56	31
YOLOv5-SB	97.91	97.22	98.47	93.67	82.50	86.78	26
YOLOv5-EB	97.72	95.87	98.74	93.65	82.70	85.49	28
YOLOv5-CBB	97.58	95.57	98.61	92.24	80.12	84.89	29

P means precision rate; R means recall; mAP means mean average precision.

**Table 6 sensors-23-01562-t006:** Performance comparison of the different algorithm models.

Models	Faster-RCNN	YOLOv3	YOLOv4	YOLOv5	YOLOv5-CB
mAP_@0.5_ (%)	92.92	88.84	92.28	94.47	98.69
Speed (fps)	11	27	21	33	31
P (%)	76.98	90.48	93.50	92.72	97.88
R (%)	93.36	76.24	79.74	92.86	96.79
F1	0.84	0.83	0.86	0.93	0.97

P means precision rate; R means recall; mAP means mean average precision.

**Table 7 sensors-23-01562-t007:** Statistics of the main reasons for detection failure.

Main Cause	Serious Stacking between Asparagus Sprouts, Leaves, and Stems	Asparagus Sprouts or Stalks Are Located at the Edge of the Image	Numerous Leaves in the Field of Vision or Background
Quantity	14	13	4

## Data Availability

Not applicable.

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
