# Peer review of "Detection of Green Asparagus in Complex Environments Based on the Improved YOLOv5 Algorithm"

_sensors, 2023, doi:10.3390/s23031562_

Round 1
Reviewer 1 Report
Interesting paper and approach. English is good and every step is well designed and analyzed. The abstract is rich of the specific Yolo family terms; I suggest to reduce the detailed terms of Yolov5 algoritm in the abstract in favour of a general presentation of the methodology. The Discussion paragraph lacks hypotheses to reduce detection failures. It would be fine to show some idea in order to resolve them: this is good to show that advancements in this work are possible yet.
Author Response
Response to Reviewer 1 Comments
Point 1: The abstract is rich of the specific Yolo family terms; I suggest to reduce the detailed terms of Yolov5 algorithm in the abstract in favor of a general presentation of the methodology.
Response 1: Thanks for your suggestion. We have revised the abstract to remove some details about the improved section in lines 15,18,22-24. The rest description of YoloV5 is easy and necessary for the work of this paper. Thanks for your understanding.
Point 2: The Discussion paragraph lacks hypotheses to reduce detection failures. It would be fine to show some idea in order to resolve them: this is good to show that advancements in this work are possible yet.
Response 2: Thanks for your constructive suggestion. As you point out, it would be fine to show some idea in order to resolve the detection failures. We also wanted to solve these problems. We did the statistics and found that the first two main causes accounted for 87 percent of the total failures. In the future work, we will make further improvements to the algorithm. For the main cause of serious stacking between asparagus sprouts, leaves and stems, we try to achieve iterative detection by changing the network structure and construct a new target detector VarifocalNet to efficiently detect dense targets. Aiming at the asparagus in the edge of image, we try to improve the feature extraction and fusion network, so that the network can learn higher level features and realize target detection through partial information. As described above, these have been amended in lines 363–374 in the revised manuscript.
All the recommendations you proposed are greatly helpful for us to polish our manuscript. We appreciate your elaborate efforts in reviewing. Thank you very much!

Reviewer 2 Report
This paper presents an improved YOLOv5 algorithm for the efficient recognition and detection of asparagus in complex environments. Although, the improved YOLOv5 algorithm shows to detect asparagus effectively in different weather and complex environments, I have the following concern.
1. This paper has many spelling and grammatical mistakes, it needs proofreading.
2. Paper Sectioning is missing. It must be added in the Introduction part of the paper.
3. More literature should be added, to show the novelty of the work.
4. Benchmarking is missing is this paper. The proposed method should be compared with other recent literature techniques. The results should be benchmarked with the recent works.
5. Section 4 (Discussion) must be improved and explained properly.
Author Response
Response to Reviewer 2 Comments
Point 1: This paper has many spelling and grammatical mistakes, it needs proofreading.
Response 1: We are sorry for the spelling mistakes and grammatical errors caused by our carelessness. In the revised version, we have made significant efforts to remove the mistakes and errors and improved writing in full text. The modified sections are indicated in the manuscript.
Point 2: Paper Sectioning is missing. It must be added in the Introduction part of the paper.
Response 2: I’m so sorry about this reply. We didn’t understand the meaning of “Paper Sectioning” even after checking the Internet. Then We compared our manuscript with the guideline and template, but we still didn’t find the question about the "Paper Section". Finally, we invited the editor to ask you about the meaning of "Paper Sectioning" by email, perhaps you are busy, we did not receive your reply. However, the paper submission deadline is up, so we submitted the manuscript first. If you feel still need to add “Paper Sectioning”, please inform the editor, and explain the meaning of “Paper Sectioning”. We will revise the manuscript in the next edition. Thanks for your understanding.
Point 3: More literature should be added, to show the novelty of the work.
Response 3: Thanks for your suggestion. A description of the difficulty of asparagus detection was added to the text, in lines 66-68. A lot of literature reviews have been added to the introduction in lines 98-106. Accordingly, the references have been modified.
Point 4: Benchmarking is missing in this paper. The proposed method should be compared with other recent literature techniques. The results should be benchmarked with the recent works.
Response 4: Thanks for your suggestion. According to your comment, we added the benchmarking and made the comparison with asparagus detection algorithms and similar techniques in recent years, in lines 346-350. Unfortunately, there is little literature on asparagus detection, so we even made the comparison with similar techniques in lines 352-354.
Point 5: Section 4 (Discussion) must be improved and explained properly.
Response 5: Thanks for your constructive suggestion. In this section, we added a comparison of asparagus detection and similar techniques in the revised manuscript. The reason of detection failure was analyzed in detail. We also gave hypotheses to reduce detection failures. It would be fine to show some idea in order to resolve them. As described in the response above, these have been amended in lines 346-350,352-354,363-374 in the revised manuscript.
All the recommendations you proposed are greatly helpful for us to polish our manuscript. We appreciate your elaborate efforts in reviewing. Thank you very much!

Reviewer 3 Report
sensors-2132382-peer-review-v1
The authors present an idea to improve the detection of asparagus plants in the filed using colour imaging coupled with deep learning. The manuscript shows a degree of novelty and can be published after taking into account the following comments.
ABSTRACT
Line 12: Change coordinate attention to Coordinate Attention
Introduction
Line 43: “Peebles et al..”. Use a consistent citation format in the whole manuscript. If the numbered citation is the formal citation format, then this should be followed. You stated 1 reference and ended the sentence with 2.
Materials and Methods
2.1 Image Data Collection: The authors should state clearly the following:
- What was the lighting conditions each day?
- How were the images acquired? S it through an automated acquisition controlled wirelessly? Or by a person?
- What are the dimensions of the field that contained the plants?
Results and Discussion
Tables 3-5: Add footnote for the symbols in the table.
The authors should put more efforts for comparing their work to the literature even for different crops.
Conclusions not Conclusion
Author Response
Response to Reviewer 3 Comments
Point 1: ABSTRACT: Line 12: Change coordinate attention to Coordinate Attention.
Response 1: We apologize for this mistake. We have changed coordinate attention to Coordinate Attention in line 14.
Point 2: Introduction : “Peebles et al..”. Use a consistent citation format in the whole manuscript. If the numbered citation is the formal citation format, then this should be followed. You stated 1 reference and ended the sentence with 2.
Response 2: We apologize for this mistake. We have changed the position of the index number [6] in lines 46. The modified sections are indicated in the manuscript.
Point 3: Image Data Collection: The authors should state clearly the following:
- What was the lighting conditions each day?
Response 3: Thanks for your suggestion. We have added the light conditions for the days of acquisition in lines 123-124.
Point 4: How were the images acquired? It through an automated acquisition controlled wirelessly? Or by a person?
Response 4: According to your comment, we have explained that the pictures were captured by a person in lines 127-128.
Point 5: What are the dimensions of the field that contained the plants?
Response 5: According to your comment, we added the size of greenhouse and explained the specific size of each row in lines 125-126.
Point 6: Tables 3-5: Add footnote for the symbols in the table.
Response 6: We are sorry for the missing footnotes caused by our carelessness. According to your comment, we have added footnote for the symbols in tables 3-5.
Point 7: The authors should put more efforts for comparing their work to the literature even for different crops.
Response 7: Thanks for your constructive suggestion. In discussion section, we added a comparison of asparagus detection and similar techniques in the revised manuscript. The reason of detection failure was analyzed in detail. We also gave hypotheses to reduce detection failures. It would be fine to show some idea in order to resolve them. As described in the response above, these have been amended in lines 346-350,352-354,363-374 in the revised manuscript.
Point 8: Conclusions not Conclusion
Response 8: According to your comment, we have revised the conclusion in the revised manuscript. First, the improvement methods and concrete measures were explained. Second, the effectiveness with distinct attention mechanisms and necks were shown. Finally, we got the conclusion. As described above, these have been revised in lines 378-380,385-391 in the revised manuscript.
All the recommendations you proposed are greatly helpful for us to polish our manuscript. We appreciate your elaborate efforts in reviewing. Thank you very much!

Round 2
Reviewer 3 Report
The authors addressed the comments and the manuscript can be published in its current form.